# New Data on the Reactions of Zirconium and Hafnium Tetrachlorides with Aliphatic Acids

**Victor D. Makhaev \***, **Larisa A. Petrova**, **Gennadii V. Shilov** and **Sergey M. Aldoshin**

Federal Research Center of Problems of Chemical Physics and Medicinal Chemistry,
Russian Academy of Sciences, Chernogolovka 142432, Russia; lapetrova@yahoo.com (L.A.P.);
genshil@icp.ac.ru (G.V.S.); aldoshin@icp.ac.ru (S.M.A.)
\* Correspondence: vim@icp.ac.ru

**Abstract:** The reaction of $ZrCl_4$ or $HfCl_4$ with excess 2-methylpropanoic acid when boiling under reflux has been studied. The formation of polynuclear Zr and Hf complexes of the composition $M_2O(i\text{-}C_3H_7CO_2)_6$ during prolonged reflux of the reaction mixtures was found. The complexes are very sensitive to hydrolysis, forming hexanuclear $[M_6(O)_4(OH)_4(i\text{-}C_3H_7CO_2)_{12}]$. The reactions have a general character for aliphatic acids and can be used as an alternative to the known methods for the synthesis of polynuclear carboxylate clusters of Group 4 metals. The crystal and molecular structures of previously undescribed $\{[Hf_6(\mu_3\text{-}O)_4(\mu_3\text{-}OH)_4(i\text{-}C_3H_7CO_2)_{12}(H_2O)]\cdot 3i\text{-}C_3H_7COOH\}$ have been determined. The molecular structure is a completely asymmetric hexanuclear cluster containing six Hf(IV) atoms united by a 4:4 $\mu_3$-O/OH system of bridges, and stabilized by twelve 2-methylpropanoate ligands, eight of which are bidentate bridging, three are chelating, and one is monodentate. The crystal structure of the complex includes three independent solvating 2-methylpropanoic acid molecules. The obtained IR spectroscopy data make it possible to determine the type of complexes in the reaction mixture. The results of the study may be useful for improving the catalytic systems for ethylene oligomerization.

**Keywords:** zirconium and hafnium 2-methylpropanoates; synthesis; X-ray structural analysis; IR spectra; ethylene oligomerization

## 1. Introduction

Group 4 metal carboxylates have been widely used in various industries, e.g., commercial ethylene oligomerization [1,2], production of high-tech ceramics [3], and nanomaterials [4]. They are promising precursors of materials for electronics [5], medicine [6,7], hydrogen energy [8,9], as metal–organic frameworks with controllable porosity [10,11], and in many other fields [12]. Both mononuclear and various polynuclear zirconium carboxylates are known [12–18]. The practical importance of Group 4 metal carboxylates has given rise to considerable interest in the study of their properties, structure, and reactivity. Zirconium tetra carboxylates, preferably, $Zr(i\text{-}C_3H_7COO)_4$, turned out to be very efficient key components of one of the commercial catalytic systems for ethylene oligomerization [1,2,19–22]. The catalytic activity of zirconium carboxylates was found, and scientific fundamentals of the process were worked out in the 1970s–1990s at the Institute of Chemical Physics Problems of the Russian Academy of Sciences [23]. In 1998, Linde (Munich, Germany) acquired the exclusive right for this process and developed, in cooperation with SABIC (Saudi Basic Industries Corporation, Riyadh, Saudi Arabia) their up-to-date commercial $\alpha$-Sablin (SABIC + Linde) process of ethylene oligomerization [1].

At present, mainly the reactions of the metal chlorides with excess corresponding organic acid are used for the synthesis of zirconium and hafnium tetra carboxylates [12,13]. Surprisingly, the effect of the synthesis conditions (time, temperature, solvent) on the reaction of $ZrCl_4$ with aliphatic acids has not been described in the literature. When

studying the reactions of $ZrCl_4$ with pivalic and 2,2-dimethylbutanoic acids, we showed that during prolonged reflux, the main reaction products are not the target tetra carboxylates $Zr(RCO_2)_4$, but polynuclear complexes of the $Zr_4O_2(O_2CR)_{12}$ and $Zr_6(O)_4(OH)_4(O_2CR)_{12}$ composition [24,25]. Over the past 30 years, a large number of polynuclear zirconium and hafnium clusters have been obtained and studied [12–18], which facilitates our work on studying the composition and structure of reaction products in $MCl_4$—organic acid systems (M = Zr, Hf), and the dependence of the products on reaction conditions.

The aim of this work was to study the effect of the reaction conditions on the composition of carboxylates formed in the $MCl_4$—2-methylpropanoic acid system (M = Zr, Hf), and to reveal the effect of acid alkyl substituent on the products by comparing the results with previously obtained data for reactions with pivalic and 2,2-dimethylbutanoic acids.

## 2. Materials and Methods

### 2.1. Starting Materials and Methods

Operations with substances sensitive to hydrolysis, carrying out reactions, and preparation of samples for instrumental methods of analysis were carried out in an inert atmosphere. $ZrCl_4$, $HfCl_4$ (Aldrich, Saint Louis, MO, USA, 99.99% purity) were used as received. 2-methylpropanoic acid (Aldrich, Saint Louis, MO, USA, 99% purity) was additionally purified by fractional distillation; hexane, *o*-xylene, and toluene (reagent grade) were distilled over lithium aluminum hydride. The reaction products were identified according to the data of chemical analysis and instrumental analysis techniques. IR spectra were recorded on PerkinElmer Spectrum 100 (PerkinElmer, Waltham, MA, USA) and Bruker Vertex 70v (Bruker Corporation, Billerica, MA, USA) spectrophotometers. The electrical conductivity of the samples was determined by the method of electrochemical impedance on a Z-500PX impedance meter (Elins, Moscow, Russia), using a symmetric cell with titanium electrodes.

An X-ray diffraction experiment was carried out on an XCalibur CCD diffractometer (Agilent Technologies, Santa Clara, CA, USA) with an EOS detector (Agilent Technologies, Santa Clara, CA, USA). Data collection, processing, determination, and refinement of the unit cell parameters were performed using the CrysAlis PRO program, version 1.171.36.20 [26]. The experiment was carried out at a temperature of 100 K. The structure was solved with a direct method. The positions and temperature parameters of the atoms were refined in the isotropic, and then in the anisotropic approximation using the full-matrix least squares method. Hydrogen atoms were calculated geometrically and refined in the rider model. Hydrogen atoms of coordinated water molecule and bridging OH groups were revealed from difference synthesis and refined with restrictions on bond lengths and thermal parameters. All calculations were performed with the SHELXL-2017/1 program package [27]. The X-ray crystal structure data have been deposited with the Cambridge Crystallographic Data Center, CCDC reference code 2172228.

### 2.2. Synthesis

#### 2.2.1. Reaction of $ZrCl_4$ with Excess 2-Methylpropanoic Acid

A mixture of 1.33 g (5.69 mmol) $ZrCl_4$ and 31 g (~350 mmol) 2-methylpropanoic acid (bp 153 °C) was refluxed in an argon atmosphere for 20 h. Excess acid was distilled off in vacuo upon heating to 120 °C. The solid residue after distillation was extracted with hexane, and the extract was filtered. Hexane was distilled off from the filtrate, and the residue was dried under vacuum at 25–150 °C for 2 h. Following this, 1.48 g of a light gray product was obtained, the composition of which corresponds to the formula $Zr_2O(C_4H_7O_2)_6$ (compound **I**). Yield: 72%. Found, %: C 39.81; H 5.89; Zr 25.5. Calculated for $C_{24}H_{42}O_{13}Zr_2$, %: C 39.98; H 5.87; Zr 25.3.

A weighed portion of compound **I** (0.426 g) was dissolved in 3 mL toluene and left in air for 4 days. The formed fine-crystalline precipitate was filtered off, dried under vacuum at 25–150 °C for 2 h. A white substance (0.225 g, 88% yield), the composition of which corresponds to the formula $Zr_6(O)_4(OH)_4(C_4H_7O_2)_{12}$ (compound **II**), was obtained. Found, %: C 33.87; H 5.07; Zr 31.57. Calculated for $C_{48}H_{88}O_{32}Zr_6$, %: C 33.43; H 5.14; Zr 31.73.

2.2.2. Reaction of $HfCl_4$ with Excess 2-Methylpropanoic Acid

A mixture of 5.93 g (18.5 mmol) $HfCl_4$ and 52 g (~590 mmol) 2-methylpropanoic acid was refluxed for 20 h. The reaction mixture was worked up as described above. A white product (3.24 g, 72% yield), the composition of which corresponds to the formula $Hf_2O(C_4H_7O_2)_6$ (compound **III**), was obtained. Found, %: C 32.61; H 5.14; Hf 39.51. Calculated for $C_{24}H_{42}O_{13}Hf_2$, %: C 32.19; H 4.73; Hf 39.86.

A weighed portion of compound **III** (0.284 g) was dissolved in 5 mL toluene and left in air for 5 days. Crystals of $[Hf_6(\mu_3-O)_4(\mu_3-OH)_4(i-C_3H_7CO_2)_{12}(H_2O)]\cdot3$ $i-C_3H_7COOH$ (compound **IV**) for the X-ray diffraction experiment were taken from the obtained mixture. Found, %: C 29.71; H 4.60; Hf 41.45. For $C_{60}H_{114}Hf_6O_{39}$ calculated, %: C 28.48; H 4.54; Hf 42.32.

**3. Results and Discussion**

*3.1. Synthesis*

In accordance with our earlier data [24,25], $ZrCl_4$ or $HfCl_4$ were refluxed with excess 2-methylpropanoic acid for 20 h to complete the reaction. As a result, chlorine-free products that are soluble in non-polar solvents—whose composition corresponds to the formula $M_2O(i-C_4H_7O_2)_6$ (M = Zr, Hf)—were obtained. Compounds of general formula $M_2O(RCO_2)_6$, (M = Zr, Hf) were described previously for pivalic, 2,2-dimethylbutanoic, methacrylic, and other acids [15,17,18,24]. For derivatives of methacrylic and 2-methylpropanoic acids of the formal composition $M_2O(RCO_2)_6$ (M = Zr, Hf), the crystal and molecular structures were determined, and it was shown that in the crystalline state they are tetranuclear clusters $[M_4O_2(RCO_2)_{12}]$ [15,17,18]. We have not yet succeeded in obtaining crystals **I** and **III** from our reaction for X-ray structural analysis; however, in accordance with the literature data, we assume that the obtained compounds **I** and **III** in the solid-state are also tetranuclear clusters $[M_4O_2(i-C_4H_7O_2)_{12}]$. A decrease in steric hindrance by replacing pivalate and 2,2-dimethylbutanoate with the less bulky 2-methylpropanoate does not suppress the condensation reaction of mononuclear $M(RCO_2)_4$ (M = Zr, Hf) to form polynuclear clusters. Consequently, the reaction has a rather general character and can be used as an alternative to the known methods for the synthesis of polynuclear clusters of Group 4 metals [14,15].

It is known that zirconium and hafnium carboxylates undergo hydrolysis very easily [13,28,29], which results in the formation of $M_6(O)_4(OH)_4(RCO_2)_{12}$ (M = Zr, Hf) clusters in the presence of water traces in the reaction mixtures. Taking into account the results of this and earlier works [14–18,24,25], the processes occurring in the systems under study can be represented by the following scheme:

$$MCl_4 + i-C_3H_7COOH \text{ (excess)} \rightarrow M(i-C_3H_7CO_2)_4 + HCl \qquad (1)$$

$$4\,M(i-C_3H_7CO_2)_4 \rightarrow 2\,M_2O(i-C_3H_7CO_2)_6 \rightarrow M_4O_2(i-C_3H_7CO_2)_{12} \qquad (2)$$

$$M(i-C_3H_7CO_2)_4 / M_4O_2(i-C_3H_7CO_2)_{12} + H_2O \rightarrow M_6(O)_4(OH)_4(i-C_3H_7CO_2) \qquad (3)$$

$$M_6(O)_4(OH)_4(i-C_3H_7CO_2)_{12} + H_2O \rightarrow [M_6(O)_4(OH)_4(i-C_3H_7CO_2)_{12}(H_2O)]_2 \qquad (4)$$

The formation of a complex mixture of products instead of the expected zirconium tetra carboxylate $Zr(RCO_2)_4$ in the reaction of $ZrCl_4$ with aliphatic acids [13] is of particular interest due to the use of the products as components of one of the commercial catalytic systems for ethylene oligomerization [1,2]. The presence of a complex mixture of carboxylates in such systems can lead to the formation of several types of coordination sites, which may affect the composition of the oligomeric fraction and formation of by-products (polyethylene) [2].

Recrystallization of $M_2O(i-C_4H_7O_2)_6$ from toluene in the presence of air followed by vacuum drying under heating results in the loss of the solvating ligands and the formation of powders whose chemical analysis data correspond to the formulas of known compounds $M_6(O)_4(OH)_4(RCO_2)_{12}$ [14,16].

In contrast to the synthesis of zirconium carboxylates by the reaction of zirconium alkoxides with acids $Zr(OR^1)_4 + R^2COOH$, which allows one to obtain only polynuclear carboxylates [12–18], the reaction in the $ZrCl_4$—RCOOH system allows one to obtain not only polynuclear carboxylates, but also the most important for some purposes zirconium tetra carboxylates $Zr(RCO_2)_4$ [20,21]. However, this reaction has a number of disadvantages, such as evolution of highly corrosive HCl and possible formation of complex mixtures of products depending on reaction conditions. Research to improve the process for obtaining $Zr(RCO_2)_4$ continues [30,31]. To overcome the disadvantageous characteristics of the currently used production process, it was proposed to react $ZrCl_4$ in an apolar solvent with the anhydride of an organic acid. The sole products of this reaction are slightly corrosive Zr tetra carboxylate and an organic acid chloride, which is a valuable basic chemical in the chemicals and pharmaceuticals industry [30].

The solid-state mechanochemical synthesis of zirconium carboxylates from $ZrCl_4$ and the corresponding sodium carboxylate is very attractive from the green chemistry viewpoint. As a result of the solid-state reaction, only zirconium tetra carboxylate is formed. This is probably due to the hindered mobility of reaction products in the solid-state, which prevents the formation of polynuclear complexes [32].

### 3.2. X-ray Crystallographic Structure

As a result of the performed X-ray diffraction studies, the molecular and crystal structure of $[Hf_6(\mu_3\text{-}O)_4(\mu_3\text{-}OH)_4(i\text{-}C_3H_7CO_2)_{12}(H_2O)]\cdot3i\text{-}C_3H_7COOH$ was determined. The molecular structure is a completely asymmetric hexanuclear cluster containing six Hf (IV) atoms united by a 4:4 $\mu_3$-O/OH system of bridges and stabilized by eight bridging 2-methylpropanoate ligands. In addition, three Hf atoms are chelated, each with one 2-methylpropanoate ligand; one Hf atom is monodentate-bonded to a 2-methylpropanoate anion, and one Hf is bonded to a water molecule. Three independent 2-methylpropanoic acid molecules have been identified in the crystal structure. The coordination water molecule, participating in hydrogen bonds with the oxygen atom of the monodentate 2-methylpropanoate ligand both inside the molecule and from the neighboring molecule, forms dimers of hexanuclear clusters $\{[Hf_6(\mu_3\text{-}O)_4(\mu_3\text{-}OH)_4(i\text{-}C_3H_7CO_2)_{12}(H_2O)]\cdot3i\text{-}C_3H_7COOH\}_2$. The solvating acid molecules participate in hydrogen bonds with hydroxo bridging groups, forming the immediate environment of the complexes. No hydrogen bonds were found between $\{[Hf_6(\mu_3\text{-}O)_4(\mu_3\text{-}OH)_4(i\text{-}C_3H_7CO_2)_{12}(H_2O)]\cdot3i\text{-}C_3H_7COOH\}_2$ dimers. Thus, dimers of hexanuclear clusters surrounded by a shell of solvating molecules are formed in the crystal structure due to hydrogen bonds. The crystal structure is stabilized mainly by van der Waals interactions, and it is formed by one-dimensional chains of cluster **IV** linked in pairs by hydrogen bonds.

Compound **IV** crystallizes in the triclinic system. The crystal structure was refined in space group P-1. A summary of data collection and single-crystal parameters is presented in Table 1. Figure 1 shows the unit cell, and Figure 2 depicts the molecular structure of compound **IV**. The asymmetric part includes six Hf atoms, twelve 2-methylpropanoate ligands, eight bridging oxygen atoms and a coordinated water molecule, as well as three crystallization molecules of 2-methylpropanoic acid.

**Table 1.** Crystal data and structure refinement for complex **IV**, $C_{60}H_{114}Hf_6O_{39}$.

| Complex | IV |
| --- | --- |
| Identification code | BM1155s_100 |
| Empirical formula | $C_{60}H_{114}Hf_6O_{39}$ |
| Formula weight | 2530.45 |
| Temperature, K | 100(1) |
| Wavelength, Å | 0.7107 |
| Crystal system, space group | Triclinic, P-1 |

**Table 1.** *Cont.*

| Complex | IV |
|---|---|
| a/Å | 12.5669(4) |
| b/Å | 15.5598(5) |
| c/Å | 22.2169(4) |
| $\alpha$/deg. | 92.465(2) |
| $\beta$/deg. | 96.440(2) |
| $\gamma$/deg. | 102.188(2) |
| Volume, Å$^3$ | 4209.6(2) |
| Z, Calculated density, Mg/m$^3$ | 2, 1.996 |
| Absorption coefficient | 7.451 mm$^{-1}$ |
| F(000) | 2436 |
| Crystal size, mm | 0.20 × 0.15 × 0.10 |
| Theta range for data collection, deg. | 2.837 to 26.569 |
| Limiting indices | $-15 \leq h < 15, -19 \leq k \leq 18, -27 \leq l \leq 27$ |
| Reflections collected/unique | 38546/17520 [R(int) = 0.0506] |
| Completeness to theta = 25.242 | 99.7% |
| Absorption correction | Semi-empirical from equivalents |
| Data/restraints/parameters | 17520/58/946 |
| Goodness-of-fit on F$^2$ | 1.033 |
| Final R indices [I > 2σ(I)] | R1 = 0.0501, wR2 = 0.1157 |
| R indices (all data) | R1 = 0.0770, wR2 = 0.1302 |
| Extinction coefficient | n/a |
| Largest diff. peak and hole, e·Å$^{-3}$ | 3.317 and $-3.214$ |

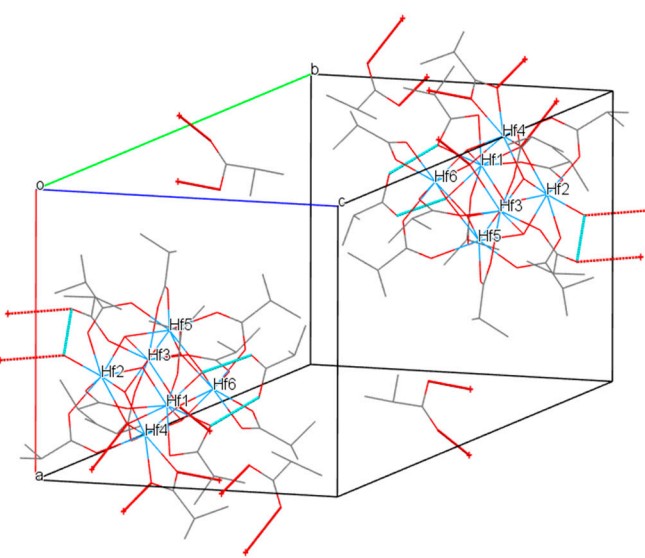

**Figure 1.** The unit cell of compound **IV**. Hafnium bonds are shown in blue, oxygen bonds in red, hydrogen bonds in aquamarine, and carbon bonds in gray.

Molecule **IV** is a hexanuclear hafnium cluster. The arrangement of Hf atoms can be represented in such a way that four of them (Hf1, Hf2, Hf3, Hf6) lie at the base, and two (Hf4, Hf5) at the vertices of a quadrilateral bipyramid (Figure 2). Bond lengths and angles in complex **IV** are listed in Tables S1 and S2 (Supplementary Materials).

Bridging oxygen atoms O1–O8 are located above the lateral triangular faces of the bipyramid, each of which connects one of the Hf atoms of the vertex with two Hf atoms of its base. Thus, each Hf atom in the molecule is bonded to neighboring Hf atoms by four $\mu_3$-bridging oxygen atoms. Charge balancing requires the presence of four OH groups.

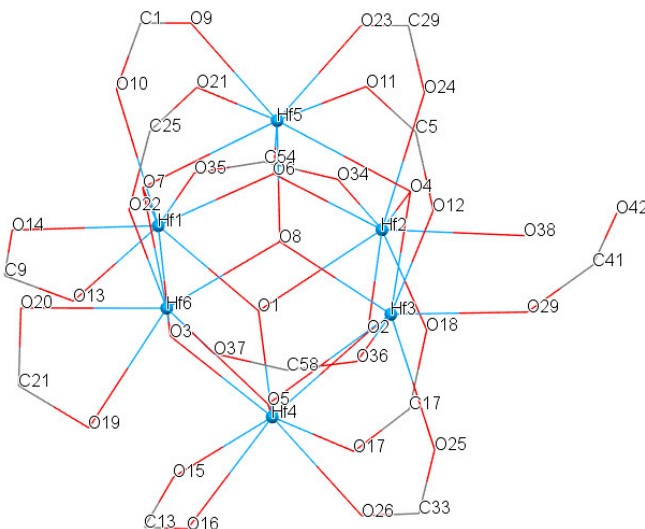

**Figure 2.** The molecular structure of compound (**IV**). For 2-methylpropanoate ligands, only the carboxyl moiety is shown. Hafnium bonds are shown in blue, oxygen bonds in red, and carbon bonds in gray.

The analysis of the bridging oxygen–metal bond lengths showed that bridging oxygen atoms are divided into the two following groups: for O1, O4, O5, and O7, the lengths of Hf–O bonds exceed 2.19 Å, and for O2, O3, O6, and O8, the lengths of Hf–O bonds are shorter than 2.10 Å. This fact allowed us to assume that the oxygen atoms of the first group are in the $OH^-$ anion, and the oxygen atoms of the second group are the $O^{2-}$ anions. A careful analysis of the electron density maps revealed differences in electron density peaks near O1, O4, O5, and O7 atoms assigned to the presence of hydrogen atoms, which confirms the correctness of the assumption. One of the reasons that in this case it was possible to determine the hydrogens near the bridging oxygens is that they are not related to each other by any symmetry elements. It is impossible, as our studies have shown, to place four bridging oxygen and OH groups within a cluster at eight positions without violating the third-order axis or the center of inversion. In our opinion, the most probable location of the bridging centers is the one identified in this work. In this case, only the inversion center is violated, and the third-order axis can be preserved.

The $OH^-$ and $O^{2-}$ anions alternately surround the Hf atoms located at the vertices. Each side of the base is connected to one of the vertices by an $OH^-$ anion, and to the other by an $O^{2-}$ anion. A similar structure is observed for other carboxylates with the $[Zr_6(\mu_3\text{-}O)_4(\mu_3\text{-}OH)_4]^{12+}$ cluster core [14,15,17].

Each Hf atom of the $[Hf_6(\mu_3\text{-}O)_4(\mu_3\text{-}OH)_4]^{12+}$ cluster core is bound to eight oxygen atoms due to coordination with 2-methylpropanoate and other ligands, but only two of them (Hf1, Hf6) have a similar ligand environment; the other four differ from each other in the type of coordination environment. Thus, the Hf5 atom located at one of the vertices, along with four $\mu_3$-O bridging ligands (O4, O6, O7, O8), is bonded to oxygen atoms (O9, O23, O11, O21) of four bridging 2-methylpropanoate ligands, connecting it to each of the bipyramid base atoms, Hf1, Hf2, Hf3, and Hf6, respectively.

The Hf4 atom located at the other vertex, along with four $\mu_3$-O bridging ligands (O1, O2, O3, O5), coordinates two oxygen atoms of two bridging 2-methylpropanoate ligands which bind it to the Hf2 and Hf3 base atoms (Hf4–O17–C17–O18–Hf2; Hf4–O26–C33–O25–Hf3), respectively. The two oxygen atoms, O15 and O16, of the chelate 2-methylpropanoate ligand result in a coordination number of eight.

The face formed by Hf1, Hf4, and Hf6 atoms differs from the other seven faces of the bipyramid in that each of the Hf atoms of this face coordinate with one non-bridging chelating 2-methylpropanoate ligand. For the other three hafnium atoms (Hf2, Hf3, Hf5),

this type of coordination was not found. The O–Hf–O angles in bonds with $\eta^2$-chelating ligands are 57.0–58.3°.

The Hf1 and Hf6 atoms are similar to each other in the type of ligand environment. Along with four bridging $\mu_3$-O atoms, each of them is bound to one $\eta^2$-chelating 2-methylpropanoate ligand (Hf1 O13, O14; Hf6 O19, O20) and one bridging 2-methylpropanoate ligand in the equatorial plane (Hf1–O35–C54–O34–Hf2; Hf6–O37–C58–O36–Hf3). For Hf1 and Hf6, the coordination number of eight is achieved by involving O atoms of the bridging 2-methylpropanoate ligands which combine Hf1or Hf6 with the Hf5 vertex. Neither Hf1 nor Hf6 are bonded with the Hf4 vertex by a bridging 2-methylpropanoate ligand. The difference in the coordination environment of Hf1 and Hf6 is that Hf1 is bound in the equatorial plane by a 2-methylpropanoate ligand to Hf2, and Hf6 to Hf3.

The coordination environment of Hf2 and Hf3 differs significantly from the hafnium atoms considered above. Along with four $\mu_3$-O bridging oxygen atoms (Hf2—O1, O2, O4, O6; Hf3—O2, O4, O5, O8), each of these centers coordinate the oxygen atom O18 or O25 of the bridging 2-methylpropanoate ligands, which combine, respectively, with Hf2 or Hf3 and the Hf4 vertex, and the O24 and O12 atoms of two bridging ligands which combine with Hf2 or Hf3 and the Hf5 vertex. In the equatorial plane, Hf2 and Hf3 are bound, respectively, to Hf1 and Hf6 by the oxygen atoms O34 and O36 of the bridging 2-methylpropanoate ligands. A distinctive feature of Hf2 is that the coordination number of eight is achieved due to the O38 atom of the water molecule. The Hf3 atom differs from other Hf atoms in the cluster by the presence of a monodentate 2-methylpropanoate ligand (Hf3–O29 bond). The monodentate 2-methylpropanoate ligand (carboxylate group O29–C41 O42) is hydrogen-bonded (O38...O42) with a water molecule solvating the Hf2 atom (bipyramid base), and with an adjacent unit cell (O42...O38'). The second hydrogen atom of the water molecule makes a bond with the neighboring unit cell (O38...O42').

Thus, the complex contains one monodentate and three non-bridging chelate 2-methylpropanoate ligands, two bridging 2-methylpropanoate ligands in the equatorial plane, and six bridging 2-methylpropanoate ligands connecting the vertices with the base atoms of the bipyramid. The bond lengths in the coordinating carboxylate groups O–C–O have the same values within three errors, and average 1.26 Å. In each carboxyl group of 2-methylpropanoic acid molecules solvating the complex, the lengths of two C–O bonds differ markedly; one of them is close to 1.23 Å, and the second to ~ 1.28 Å (Table S1). The difference in the lengths of C–O bonds suggests that short bonds correspond to C=O, and longer bonds correspond to C–O–H groups of 2-methylpropanoic acid molecules, which agrees with the data for aliphatic acids [33–35]. However, it was not possible to reveal the positions of hydrogen atoms from difference syntheses for the acid molecules.

In the crystal structure of **IV**, hydrogen bonds that arise between the coordination water molecule O38 and the oxygen atom O42 of the monodentate 2-methylpropanoate ligand both inside the Hf hexanuclear cluster and with the neighboring cluster are observed, forming pairs of hexanuclear clusters (Figure 3). Hydrogen bonds are also observed between $\mu_3$-OH groups and oxygen atoms of solvating 2-methylpropanoic acid molecules. The parameters of the considered hydrogen bonds are listed in Table 2. Only oxygen atoms of the solvating molecules, which have short C=O bonds, are involved in hydrogen bonds with the $\mu_3$–OH groups of the $[Hf_6(\mu_3\text{-}O)_4(\mu_3\text{-}OH)_4]^{12+}$ cluster core. However, the oxygen atoms of the supposed C–O–H groups have short contacts with oxygen atoms from the coordination environment. Thus, O28...O16 has a length of 2.640 Å; O31...O14 has a length of 2.679 Å; and O33...O15 has a length of 2.719 Å. These contacts can only be explained by the formation of hydrogen bonds. Note that oxygen atoms O14, O15, and O16 belong to two non-bridging $\eta^2$-chelate 2-methylpropanoate ligands. We calculated the positions of hydrogen atoms at the O28, O31, and O33 atoms, after which they were refined in the rider scheme (See Materials and Methods).

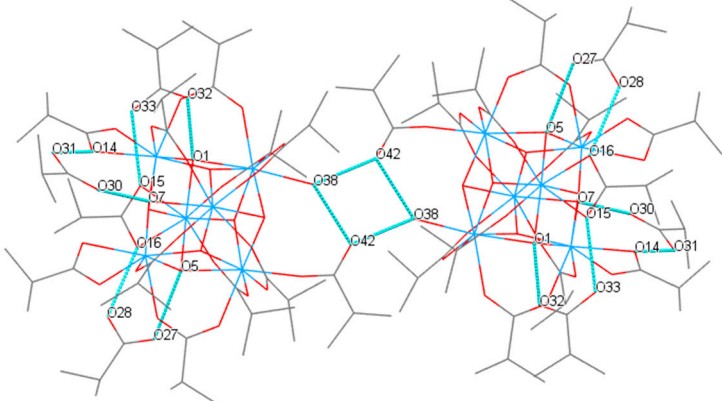

**Figure 3.** Hydrogen bonds between hexanuclear Hf clusters in the crystal structure of complex **IV**. Hafnium bonds are shown in blue, oxygen bonds in red, carbon bonds in gray, and hydrogen bonds, in aquamarine.

**Table 2.** Parameters of the hydrogen bonds and weak contacts in compound (**IV**).

| D–H | d(D–H, Å) | d(H...A, Å) | <DHA | d(D...A, Å) | A |
|---|---|---|---|---|---|
| O5–H5 | 0.854 | 1.982 | 169.91 | 2.827 | O27 |
| O7–H7 | 0.893 | 1.871 | 176.30 | 2.762 | O30 |
| O38–H38C | 0.959 | 1.776 | 166.12 | 2.717 | O42 |
| O38–H38D | 0.958 | 1.840 | 158.24 | 2.753 | O42 * |
| O28–H28 | 0.840 | 1.806 | 171.78 | 2.640 | O16 |
| O31–H31 | 0.840 | 1.847 | 170.07 | 2.679 | O14 |
| O32–H32 | 0.840 | 2.023 | 150.72 | 2.786 | O1 |
| O33–H33 | 0.840 | 1.884 | 172.26 | 2.719 | O15 |

Note: * [ −x + 1, −y, −z].

All hydrogen bonds are formed between molecules within one-dimensional chains. Hydrogen bonds between 1D chains were not found (Figure 4).

A similar Zr compound was obtained and studied earlier by the reaction of $Zr(OBu)_4$ with 2-methylpropanoic acid [17]. The structures of zirconium [17] and hafnium (this work) hexanuclear 2-methylpropanoate clusters are similar. Powder diffraction patterns of Zr and Hf clusters are not identical due to the difference in the metrics (periods and angles of the unit cells) of the crystals. The authors [17] indicated that the hydrogen atoms of the hydroxo groups were calculated geometrically. In our work, we succeeded in identifying hydrogen atoms of the hydroxo groups from difference syntheses.

For a number of $[Zr_6(\mu_3\text{-}O)_4(\mu_3\text{-}OH)_4(RCOO)_{12}]$ clusters with various carboxylate ligands (pivalate, R = $–C(CH_3)_3$; 2,2-dimethylbutanoate, R = $–C(CH_3)_2C_2H_5$ [14]; methacrylate, R = $–C(CH_3)=CH_2$) [15]; acetate, R = $–CH_3$; propionate, R = $–CH_2CH_3$ [16]; and 2-methylpropanoate, R = $–CH(CH_3)_2$ [17]), the crystal and molecular structures were determined by X-ray diffraction analysis. It was found that the clusters with bulky alkanoate moieties (pivalate, 2,2-dimethylbutanoate) have high symmetry. According to the authors [14], the structure of these compounds consists of hexanuclear clusters $[Zr_6(O)_8]^{12+}$, in which six equivalent metal ions are located at the vertices of the octahedron. They did not succeed in experimentally revealing the difference between $\mu_3$-O and $\mu_3$-OH groups. Each of the zirconium atoms in the cluster is bonded to four neighboring zirconium atoms by bidentate bridging carboxylate groups. No other types of coordination of carboxylate groups were found. The methacrylate, acetate, propionate, and 2-methylpropanoate clusters were found to contain two types of carboxylate ligands, namely, nine bridging and three non-bridging $\eta^2$-chelates [15,17]. A possible reason for the change in the structure of the clusters in question is the difference in the steric properties of hydrocarbon moieties of carboxylate ligands. The influence of electronic and steric factors on the structure, prop-

erties, and catalytic activity of both mononuclear and polynuclear Zr and Hf complexes requires further study.

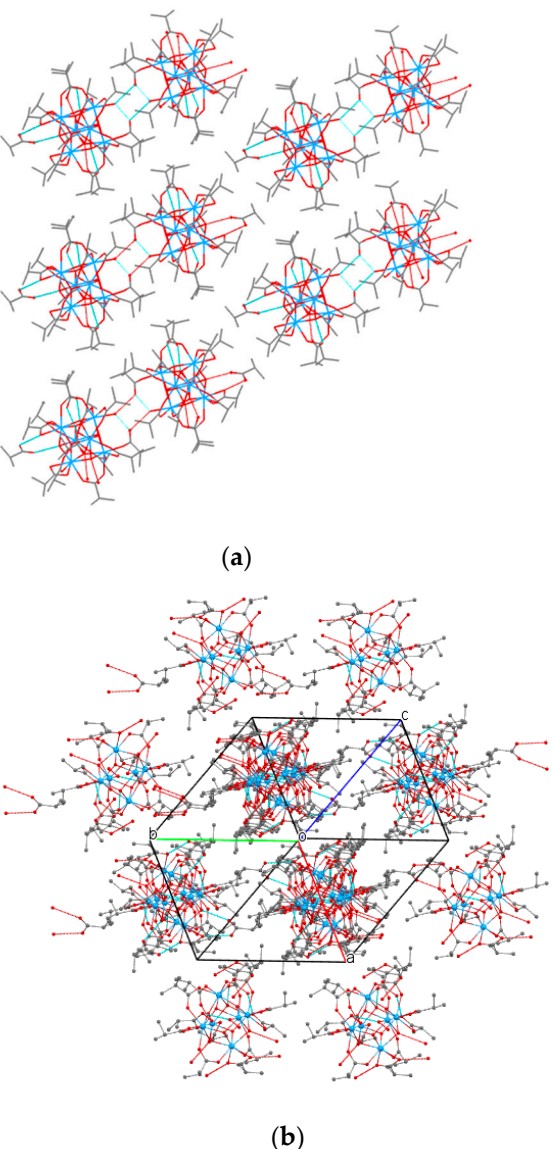

(**a**)

(**b**)

**Figure 4.** Crystal packing of **IV**: (**a**) view down crystallographic b axis; (**b**) view showing one-dimensional chains separated by hydrocarbon moieties. Hafnium bonds are shown in blue, oxygen bonds in red, hydrogen bonds in aquamarine, and carbon bonds in gray.

A large difference in the peak and hole was found for cluster **IV**. There are many reasons for a higher than expected residual density maximum outside of metal atom locations, such as errors in operating the diffractometer, the data processing software or the refinement software, temperature fluctuations at the detector, cosmic rays, natural radioactivity, errors in absorption and extinction corrections, thermal diffuse scattering, radiation damage of the crystal, wrong scattering factors—e.g., non-relativistic for heavy atoms, crystal chemical instability, twinning, wrongly assigned atom types, disorder when the molecule under consideration is large and has bulky groups, etc. [36]. Comparison of our data with the literature results [14,16] shows that our data on the largest peak-hole difference are consistent with published data for similar clusters.

Investigation of crystal **IV** by the method of impedance measurement showed the absence of electrical conductivity, which corresponds to the absence of a system of conjugated

hydrogen bonds in the crystal structure. More deeply hydrolyzed products are likely to be electrically conductive.

*3.3. IR Spectra*

Table 3 shows the IR spectroscopic data for compounds **I–IV**. The assignment of some absorption bands is given on the basis of the literature data [13,14,37,38]. Unfortunately, there are no data in the literature on the IR spectroscopy of the zirconium analog of complex **IV** to compare with our results.

**Table 3.** IR spectra of compounds **I–IV** in the range 4000–300 cm$^{-1}$ ($\nu$, cm$^{-1}$).

| Assignment | 2-Methylpropanoic Acid | I | II | III | IV |
|---|---|---|---|---|---|
| $\nu$(OH), $\mu_3$-OH groups | | | 3676 w, sharp | | 3657 w, sharp |
| $\nu$(OH) acid | 3500–3200 wide | | | | |
| $\nu$(CH$_3$) | 2979 s, 2968 m, 2875 m | 2966, 2923, 2875 | 2966, 2923, 2875 | 2966, 2923, 2875 | 2966, 2923, 2875 |
| $\nu$(COO) acid | 1709 vs | | 1717 m | | 1700 m |
| $\nu_{as}$(COO) | 1572 s, 1556 vs | 1589 s, wide | 1612 m, 1542 s, 1524 s | 1593 s, 1561 s | 1604 m, 1547 s, 1530 s |
| $\delta_{as}$(CH$_3$) | 1478 s | 1474 s | 1471 s | 1473 s | 1473 s |
| $\nu_s$(COO) | 1418 m | 1433 s | 1431 s | 1430 s | 1434 s |
| $\delta_s$(CH$_3$) | 1387 s, 1368 s, 1334 s | 1377, 1363 m | 1362 m | 1376 m | 1378 w, 1363 w |
| $\nu$(C–(CH$_3$)$_2$), $\delta$(C–C) | 1290 s | 1297 s, 1214 m | 1296 m, 1213 m | 1303 w, 1213 m, 1167 w | 1296 m, 1215 w, 1187 w |
| $\nu$(C–O) acid | 1241 vs, 1169 m | | | | |
| $\rho$(CH$_3$) | 1100 m, 1080 m | 1097 m, 965 w | 1097 m, 964 w | 1097 m, 986 vw | 1096 m |
| $\nu$(C–C) | 937 s | 934 w | 932 vw | 931 vw | 935 vw |
| | 811 m | 861 m, 771 m | 842 w, 766 w | 865 w, 841 w, 764 w | 863 w, 771 w |
| $\delta$(OCO) | 630 m | | 650 w, wide | 657 w, wide | 657 w, wide |
| $\pi$(OCO) out-of-plane | | 605 m, 560 w | 596 w | 570 vw | 568 w, wide |
| | 544 w, 524 w | | 541 m | 547 vw | 521 m |
| $\pi$(OCO) in-plane | | 464 s, wide 435 s, wide | 488 m, wide, 421 s, wide | 492 vw, 452 vw | |
| | | | 379 w | 351 m, wide | 372 w |

In the spectra of compounds **I** and **III**, obtained by prolonged reflux of ZrCl$_4$ with 2-methylpropanoic acid, there is no intense absorption band $\nu_{as}$(COO) at ~1630 cm$^{-1}$ typical of zirconium tetra carboxylates [13]. Instead, an intense broad absorption band was found at ~1590 cm$^{-1}$, which is characteristic of the previously obtained derivatives of pivalic and 2,2-dimethylbutanoic acids of the composition Zr$_2$O(RCO$_2$)$_6$ [24,25]. The absorption band at 1434 cm$^{-1}$, as for the previously studied compounds, can be assigned to the $\nu_s$(COO) vibrations.

Complexes **II** and **IV** are formed as a result of partial hydrolysis of compounds **I** and **III**. As noted above, there are 2-methylpropanoate ligands of several coordination modes (monodentate, bidentate chelating, and bidentate bridging ones), as well as solvating 2-methylpropanoic acid and water molecules that are hydrogen-bonded with other ligands in the [Hf$_6$($\mu_3$-O)$_4$($\mu_3$-OH)$_4$(*i*-C$_3$H$_7$CO$_2$)$_{12}$(H$_2$O)]·3*i*-C$_3$H$_7$COOH] (IV) structure. This results in a large difference in the IR spectra of 2-methylpropanoate complexes **II** and **IV** and those for the spectra of hexanuclear derivatives of more sterically hindered pivalic and 2,2-dimethylbutanoic acids, in which only carboxylate ligands with bidentate bridging coordination are present [14,25].

The $\nu_{as}(COO)$ vibrations in **IV** include absorption bands at 1604 (monodentate carboxylate group [37,38]), 1547, and 1530 cm$^{-1}$. The absorption band at 1434 cm$^{-1}$, as for the previously studied compounds, can be assigned to the $\nu_s(COO)$ vibrations. A similar IR spectrum is observed for the zirconium analog (compound **II**). After removal of the solvating acid in vacuum, the shape of the spectrum in the range of 1650–300 cm$^{-1}$ nearly does not change.

The literature and data obtained in this work show that different types of zirconium carboxylates differ markedly in their IR spectra. Mononuclear carboxylates $Zr(RCO_2)_4$ are characterized by the presence of an intense sharp absorption band at ~1630 cm$^{-1}$, which indicates the equivalence of all four carboxyl groups [13]. The presence of $\mu_3$-OH groups in hexanuclear $Zr_6(O)_4(OH)_4(RCO_2)_{12}$ clusters results in the appearance of a sharp, weak absorption band at ~3400–3670 cm$^{-1}$, typical of hexanuclear clusters [14,25]. These absorption bands are absent in the IR spectra of tetranuclear complexes. Thus, IR spectroscopy data make it possible to clearly distinguish between mononuclear $Zr(RCO_2)_4$ carboxylates, tetranuclear $Zr_4O_2(RCO_2)_{12}$, and hexanuclear $Zr_6(O)_4(OH)_4(RCO_2)_{12}$ clusters.

## 4. Conclusions

As a result of this and previous studies, the formation of polynuclear complexes of the $M_2O(RCO_2)_6$ composition upon prolonged reflux of $MCl_4$ (M = Zr, Hf) in excess aliphatic (pivalic, 2,2-dimethylbutanoic, and 2-methylpropanoic) acid was found. The complexes are very sensitive to hydrolysis, forming hexanuclear $[M_6(O)_4(OH)_4(RCO_2)_{12}]$. The reactions have a rather general character for aliphatic acids, and can be used as an alternative to the known methods for the synthesis of polynuclear carboxylate clusters of Group 4 metals. The crystal and molecular structures of the previously undescribed complex $[Hf_6(\mu_3\text{-}O)_4(\mu_3\text{-}OH)_4(i\text{-}C_3H_7CO_2)_{12}(H_2O)]\cdot3i\text{-}C_3H_7COOH]$ were determined. The molecular structure is a completely asymmetric hexanuclear cluster containing six Hf(IV) atoms united by a 4:4 $\mu_3$-O/OH bridges system and stabilized by eight bridging, three non-bridging chelate, and one monodentate 2-methylpropanoate ligand. In addition, one Hf atom coordinates the water molecule. The crystal structure is stabilized mainly by van der Waals interactions and hydrogen bonds. The IR spectroscopy data obtained by us in this work and in earlier works for $M(RCO_2)_4$, $M_2O(RCO_2)_6$, $M_6(O)_4(OH)_4(RCO_2)_{12}$ (R = $(CH_3)_2CH$–, $(CH_3)_3C$–, and $C_2H_5(CH_3)_2C$–) complexes enable one to reliably determine the presence and type of the complexes. The results of the study concerning the dependence of the product composition on the reaction conditions and on the nature of hydrocarbon moieties of carboxylate ligands may be useful for improving catalytic systems for ethylene oligomerization.

**Supplementary Materials:** The following supporting information can be downloaded at: https://www.mdpi.com/article/10.3390/compounds4020018/s1, Table S1: Bond lengths [Å] for complex **IV**; Table S2: Bond angles [deg.] for complex **IV**; Cif-file for complex **IV**; Checkcif-file for complex **IV**.

**Author Contributions:** Conceptualization, writing—review and editing, V.D.M.; investigation, writing—original draft, L.A.P.; X-ray structure investigation, writing—original draft, G.V.S.; supervision, S.M.A. All authors have read and agreed to the published version of the manuscript.

**Funding:** This study was carried out as part of a state assignment of the Federal Research Center of Problems of Chemical Physics and Medicinal Chemistry of the Russian Academy of Sciences (State registration no. 124013000692-4 and 124013100858-3; topic no. FFSG-2024-0006 and FFSG-2024-0009). This research did not receive external funding.

**Data Availability Statement:** Data are contained within the article or supplementary material.

**Acknowledgments:** The authors are grateful to L.S. Leonova (FRC PC MC RAS, Chernogolovka, Russia) for carrying out research using the impedancemetry technique. Chemical analysis, registration of diffraction patterns, and IR spectra were performed using the equipment of the Multi-User Analytical Center of FRC PCP MC RAS.

**Conflicts of Interest:** The authors declare no conflicts of interest.

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
