# Peer review of "New Data on the Reactions of Zirconium and Hafnium Tetrachlorides with Aliphatic Acids"

_compounds, doi:10.3390/compounds4020018_

Round 1

Reviewer 1 Report

Comments and Suggestions for Authors

This work synthesized Hf complex ([Hf63-O)43-OH)4(i-C3H7CO2)12(H2O)]3i-C3H7COOH). This crystal structure has been determined. The IR spectroscopy data obtained make it possible to determine the type of complex in the reaction mixture. The results of the study may be useful for improving the catalytic systems for ethylene oligomerization. This manuscript can be published after some changes. The comments are listed below.

1.     It is best to combine the title into one sentence.

2.     There are two repeated subheadings (2.1 Starting Materials and Methods and 2.1 Synthesis).

3.     More related new works are suggested to be cited, such as: Adv. Funct. Mater. 2023, DOI: 10.1002/adfm.202303833; Carbon Energy, 2023, 5, e265.

4.     The authors provided detailed descriptions of the crystal structure and FTIR data, but there are still some other characterizations lacking, such as TGA, SEM, and so on. The morphology and thermal stability of the complex also need to be understood.

5.     The authors said “The results of the study may be useful for improving the catalytic systems for ethylene oligomerization”. Have the authors attempted to conduct the catalytic performance measurements of ethylene oligomerization?

Author Response

Thank you for your review and comments.

  1. Combining the title into one sentence - if the editors deem it necessary, I agree.
  2. - subheadings, corrected.
  3. More related new works are suggested to be cited - the suggested references refer to MOFs. Our work is related to catalysts for oligomerization processes.
  4. There are still some other characterizations lacking, such as TGA, SEM, and so on. The morphology and thermal stability of the complex also need to be understood. - these properties will be studied further for most high-performance samples. We are now obtaining new clusters to study their catalytic activity.
  5. Have the authors attempted to conduct the catalytic performance measurements of ethylene oligomerization? - Yes, we did it. Preliminary data showed good results. We cannot disclose details yet.

Reviewer 2 Report

Comments and Suggestions for Authors

Numerous crystal structures of Zr6O4((OH)4(carboxylate)12 clusters have been extensively documented in the literature, as evidenced by references 14, 15, and 17. In this manuscript, authors delved into the X-ray crystallography determination of the structural features of their Hafnium (Hf) analogues, spanning five pages of detailed description, constituting nearly half of the manuscript. Notably, it is worth mentioning that a similar hexanuclear Hf cluster with methacrylate ligands, in place of isobutyrate, was reported approximately two decades ago in "Monatshefte fur Chemie, 2003, 134, 1053. Regrettably, this pertinent reference was not cited within the present manuscript. Moreover, the synthetic pathway employed to obtain the Hafnium cluster compound follows a conventional route. Regrettably, the structural characterization of the intermediate compound Hf2O(C4H7O2)6 remains inadequately addressed. In light of this, one may question whether merely presenting the structure of such a commonplace complex qualifies as a scientific paper in today's research landscape.

Author Response

Thanks for the important reference on the structure of Hf methacrylate. It is included in the list of references. Methacrylates are important for the production of hybrid materials. MOFs based on Zr6 clusters are being actively researched.

However, our research has a completely different goal, the search for effective precursors of Zr and Hf catalysts for the production of polyalpha-olefins. Zirconium alkanoates, in particular isobutyrates, at present are considered to be the most efficient in this process, but no systematic research has been carried out in this area.

There is extremely little reliable data on the composition, structure and properties of zirconium derivatives of fatty acids. Therefore, to search for new effective catalytic systems, in many cases various carboxylates of uncertain composition and structure (zirconium soaps) are used. In fact, this is a random search. To achieve our goal, we intend to study the dependence of the efficiency of precursors on the structure of mononuclear, tetranuclear, hexanuclear Zr clusters, and on the nature of carboxylate groups. These data are not available in the literature.

We are currently synthesizing and studying a range of mono- and polynuclear Zr and Hf carboxylates. Preliminary catalytic results show the effectiveness of this approach. For those involved in improving the oligomerization process based on Zr carboxylates (SABIC, SIBUR, etc.), the presented results will be of significant interest.

In addition, the reaction we used (MCl4 + RCOOH), in contrast to the literature one (M(OBu)4 + RCOOH), makes possible to obtain all of the indicated types of clusters and shows the step-by-step transformation of mononuclear compounds into tetra- and then hexanuclear ones depending on the reaction conditions. Along with our previous works, it complements the existing limited data on the interaction of ZrCl4 with aliphatic acids.

A careful consideration of the structure of the hexanuclear Hf isobutyrate we obtained is caused with the presence in it of five different types of ligand environments of Hf atoms, which can lead to the formation of different types of catalytic sites.

Reviewer 3 Report

Comments and Suggestions for Authors

This structure has the advantage of being lower symmetry and so the OH and O atoms are distinguishable unlike the similar referred to structure in ref 17 so adds value to scientific knowledge

No structure factors were supplied with CIF in deposited data .  So I contacted CCDC for the deposited CIF and Checkcif and found  supplied by the Journal  was not complete - omitted level A alerts found with CIF obtained from CCDC  and the CCDC checkcif  said it contained Afix 1 (meaning parameters were not refined - this may have taken out the alert A) . The text states Hydrogen   atoms of coordinated water molecule and bridging OH groups were revealed from difference synthesis and refined with restrictions on bond lengths and thermal parameters.”

 these are usually bond length restraints and displacement parameters riding on the bound atom what was constrained with Afix 1 should be stated and the cif and check cif deposited with the SI should  be the same as that with the CCDC

However the max residual  peaks are given in the Table 1 and Hf structures would be expected to have peaks near the metal atoms . what absorption correction was carried out ? was twinning checked for ?

Packing diagram Figs 1 3 and 4

needs to be bigger, show unit cell axes  labels more clearly

 Fig 1  should  show all isobutyric acid molecules coordinated  to metal centres, they are not uncoordinated as is implied by current packing diagram . This coordination helps them to be ordered in the lattice

Comments on the Quality of English Language

could be improved but is comprehensible

Author Response

Thank you for valuable review and comments.

We corrected our manuscript according to your comments. Structural factors are now contained in the cif file.

The restriction Afix 1 was removed and the structure was refined.

The CIF file has been checked.

OH groups in the solvents were refined using a rider scheme.

The positions of the hydrogen atoms of the bridging OH groups and water have been refined with restrictions on bond lengths.

What absorption correction was carried out ? was twinning checked for? - The empirical correction for absorption is made using spherical harmonics; the crystal is not a twin. The residual electron density of Hf is at the level of the electron density of hydrogen. Experimental experience shows that to reduce this residual electron density, crystals of the smallest possible size and the best quality are needed. Unfortunately, we were unable to obtain them. As literature data show, difficulties in X-ray diffraction determination of the structure of such polynuclear complexes of heavy metals are observed quite often. We ask the respected reviewer to take into account that the main goal of our research is not X-ray diffraction studies as such, but the search for new effective catalysts for ethylene oligomerization based on instrumentally characterized complexes. In our opinion, we have completed the task, perhaps with some imprecision from a strictly crystallographic point of view.

Packing diagram Figs 1 3 and 4 needs to be bigger, show unit cell axes labels more clearly. - We made the corrections.

Fig 1 should show all isobutyric acid molecules coordinated to metal centres, they are not uncoordinated as is implied by current packing diagram . This coordination helps them to be ordered in the lattice. - We made the corrections.

Reviewer 4 Report

Comments and Suggestions for Authors

This work could be an interesting article describing the method of synthesizing zirconium and hafnium hexanuclear [M6(O)4(OH)4(i-C3H7CO2)12] compounds and then their comparison with similar compounds already published in the literature. At the moment, this is not the case. If the authors arbitrarily change the concept of this work, I will express my positive opinion. Otherwise, I will not recommend this article for publication despite the potentially interesting structural chemistry.

List of comments:

1. "Title: Interaction of Zirconium and Hafnium Tetrachlorides with Iso- 2 butyric Acid. Crystal and Molecular Structure of [Hf6(μ3-O)4(μ3-OH)4(i-C3H7CO2)12(H2O)].3i-C3H7COOH.”

The work is focused on the synthesis of oxo/hydroxo group 4 clusters by the thermal decomposition of [M(i-C3H7CO2)4] precursors and then, by hydrolysis of formed [M2O(i-C3H7CO2)6]. Group 4 tetrachlorides are used only as starting reagents. I found no results confirming interactions between MCl4 and iso- 2 butyric acid. Therefore, I suggested changing the title of this work.

2.Abstract:

a) "The interaction of ZrCl4 or HfCl4 with excess isobutyric acid when boiling under reflux has been studied."

Instead, I propose: Reactions of ZrCl4 or HfCl4 with excess isobutyric acid under reflux lead to the formation of [M2O(i-C3H7CO2)6], which undergo hydrolysis, leading to hexanuclear compounds [M6(O)4(OH)4(i-C3H7CO2)12].

b) The abstract contains many sentences that should be included in the Introduction. There is no clear description of what was done in the work.

3. Introduction:

a) The chemistry of oxo-metal carboxylates by vacuum distillation/temperature decomposition (allow 200 C) of metal carboxylates is an issue known from the beginning of the XX century. The synthesis of metal hydroxo clusters by hydrolysis of metal complex, stoichiometric addition of water, or exposition for moisture is extensively investigated in the field of sol-gel synthesis, catalysis, polymerization, and synthesis of metal-organic frameworks. For the group 4 elements (particularly for Ti), oxo-metal clusters are also readily formed by ether or alcohol decomposition.

b) The CCDC database contains 70 structures of oxo/hydroxo clusters of zirconium and 11 structures of hafnium with aliphatic carboxylate ligands, so it would have been nice to read a little more about the structural chemistry of such compounds in the Introduction.

4. Materials and Methods

a) The synthesis section should include a reference to previously published work on the synthesis of [Zr6(O)4(OH)4(i-C3H7CO2)12] compound II. Compound II, according to reference 17 was previously obtained by the reaction of [Zr(OnBu)4] with i-C3H7COOH in the form of [Zr6(O)4(OH)4(i-C3H7CO2)12(H2O)]·3 i-C3H7COOH. Comparison of the powder diffractogram of crystalline compound II with the diffractogram simulated from cif file for previously published [Zr6(O)4(OH)4(i-C3H7CO2)12(H2O)]·3i-C3H7COOH or with compound IV would confirm the final structure of compound II proposed by the Authors.

5. X-ray crystallographic structure

a) "Compound IV crystallizes in the triclinic system. The crystal structure was refined in space group P-1….and then There are two Hf6(μ3-O)4(μ3-OH)4(i-C3H7CO2)12(H2O)].3i-C3H7COOH molecules per unit cell related by the inversion center."

The last sentence should be removed for obvious reasons.

b) The figures are very difficult to read (Figure 1, Figure 2, Figure 4a).

c) Could the Authors explain the presence of Hf-Hf contacts in the figures?

d) "Based on the charge balance, there should be the charge of 12– per 8 oxygen atoms in a molecule. This is possible when four oxygen atoms have the charge of 2–, and four, of 1–. The analysis of the bridging oxygen–metal bond lengths showed that bridging oxygen atoms are divided into two groups: for O1, O4, O5, O7, the lengths of Hf–O bonds exceed 2.19 Å , and for O2, O3, O6, O8, the lengths of Hf– O bonds are shorter than 2.10 Å . This fact allowed us to assume that the oxygen atoms of the first group are in the OH– anion, and the oxygen atoms of the second group are the O2– anions. A careful analysis of the electron density maps revealed difference electron density peaks near O1, O4, O5, O7 atoms assigned to the presence of hydrogen atoms, which confirms the correctness of the assumption."

And

“So, O28…O16 has a length of 2.640 Å ; O31...O14, 2.679 Å ; O33...O15, 2.719 279 Å . These contacts can only be explained by the formation of hydrogen bonds. Note that oxygen atoms O14, O15, O16 belong to two non-bridging chelate isobutyrate ligands. We calculated the positions of hydrogen atoms at the O28, O31 and O33 atoms, after which they were refined in the rider scheme."

It is a routine procedure and is more suitable for inclusion in the experimental part.

e) The checkcif procedure performed for structure BM1155s_100 contains 2 alerts A and 19 alerts B, but the checkcif file originally applied as supplementary material for this work is without the alerts mentioned above. The structure is resolved correctly; however, absorption correction using an analytical model should be performed, and most of these alerts will probably be removed. The presented cif file is probably without absorption correction because Tmax=1.000 and this value should be different from 1.

f) The below part of the manuscript should be placed at the beginning of the crystal structure description of compound IV, starting on page 3.

"As a result of the performed X-ray diffraction studies, the molecular and crystal structure of [Hf6(μ3-O)4(μ3-OH)4(i-C3H7CO2)12(H2O)].3i-C3H7COOH crystals was determined. The molecular structure is a completely asymmetric hexanuclear cluster containing six Hf (IV) atoms united by a 4:4 μ3-O/OH system of bridges and stabilized by 8 bridging isobutyrate ligands. In addition, three Hf atoms are chelated each with one isobutyrate ligand; one Hf atom is monodentate bonded to isobutyrate anion, and one Hf is bonded to a water molecule. Three independent isobutyric acid molecules have been identified in the crystal structure. The coordination water molecule, participating in hydrogen bonds with the oxygen atom of the monodentate isobutyrate ligand both inside the molecule and from the neighboring molecule, forms dimers of hexanuclear clusters {[Hf6(μ3-O)4(μ3-OH)4(i- C3H7CO2)12(H2O)].3i-C3H7COOH]}2. The solvating molecules participate in hydrogen bonds with hydroxo bridging groups, forming the immediate environment of the complexes. No hydrogen bonds were found between {[Hf6(μ3-O)4(μ3-OH)4(i-C3H7CO2)12(H2O)].3i-C3H7COOH]}2 dimers. Thus, dimers of hexanuclear clusters surrounded by a shell of solvating molecules are formed in the crystal structure due to hydrogen bonds. The crystal structure is stabilized mainly by van der Waals interactions and is formed by one-dimensional chains of clusters IV linked in pairs by hydrogen bonds."

g) "Due to the presence of different types of M–O bonds in compounds I–IV, special studies are required for a correct assignment of the corresponding absorption bands."

The above sentence should be removed.

Author Response

We are grateful to the Reviewer for helpful comments. We agree with points 1, 2, 5 a-c, f, g. We changed the title of the work, focusing on the study of the composition of Zr carboxylates formed in the ZrCl4 – 2-methylpropanoic acid system used for the preparation of efficient commercial catalysts for ethylene oligomerization. The new title “New Data on the Reactions of Zirconium and Hafnium Tetrachlorides with Aliphatic Acids” refers to the title of a key article on the topic “A Study of the Reactions of Zirconium(IV)Chloride with Some Aliphatic Acids” by J. Ludvig and D. Schwartz (Inorg. Chem. 1970, 9, 607-611. https://doi.org/10.1021/ic50085a034. Surprisingly, studies of the actual composition of the products of such reactions are not described in the literature. In the absence of other papers on the topic it was believed that only tetracarboxylates were formed as a result of the reaction. We have shown that this is not the case.

Accordingly, we changed the abstract. Figures are corrected. Proposed changes in the text are made.

Responses to Reviewer comments.

Point 3: a) We are aware of thermal decomposition of metal carboxylates for the synthesis of metal oxo complexes, but for industrially important reactions in ZrCl4 – aliphatic acid systems this reaction was not described. The only keystone paper of Ludwig and Schwarts on this issue does not mention formation of other carboxylates except Zr(RCO2)4;

b) The presence of a large number of structurally characterized polynuclear zirconium carboxylates helps us in our research. The more substances that are characterized, the easier it is for us to identify them in the products of reactions of MCl4 with acids.

Point 4. - In our opinion, it is more appropriate to provide reference 17 in the “Results and Discussion” section, where the advantages and shortcomings of known methods for the synthesis of various types of zirconium carboxylates are discussed. Powder diffraction patterns of Zr and Hf clusters are similar, but not identical due to the difference in their metrics (periods and angles of the unit cell).

Point 5 c. - The Mercury program shows the presence of Hf-Hf contacts. For clarity, we have removed them from the Figures.

Point 5 e. The checkcif procedure performed for structure BM1155s_100 contains 2 alerts A and 19 alerts B, but the checkcif file originally applied as supplementary material for this work is without the alerts mentioned above. The structure is resolved correctly; however, absorption correction using an analytical model should be performed, and most of these alerts will probably be removed. The presented cif file is probably without absorption correction because Tmax=1.000 and this value should be different from 1.

- We solved the structure in 2018, when CCDC requirements were not as strict as they are now. Checkcif file from this time was given in Supplementary. Now, the requirements have become more stringent. The absorption correction was performed in the range (Table 1):

exptl_absorpt_correction_T_min                   0.34002

_exptl_absorpt_correction_T_max                   1.00000

_exptl_absorpt_correction_type            'multi-scan'

In the Cif-file it is said: Empirical absorption correction using spherical harmonics, implemented in SCALE3 ABSPACK scaling algorithm.

Note that we used X-ray diffraction only as an analytical tool. Taking into account the data of chemical analysis, IR spectroscopy and the close similarity of the known hexanuclear Zr 2-methylpropionate cluster and the considered Hf complex, we can say that we have determined the structure of the Hf cluster with an accuracy satisfactory for our purposes. Difficulties in determining the structure are associated with the imperfection of the crystal and its tendency to crack. Our attempts to obtain a more perfect crystal have not yet led to success.

Reviewer 5 Report

Comments and Suggestions for Authors

The paper details the formation of previously unreported polynuclear Zr and Hf complexes with the composition M2O(i-C3H7CO2)6 during prolonged reflux of the reaction mixtures. Furthermore, the crystal and molecular structure of the hitherto undescribed {[Hf6(μ3-O)4(μ3-OH)4(i-C3H7CO2)12(H2O)*3i-C3H7COOH} has been determined. Additionally, the IR spectroscopy data obtained make it possible to determine the type of complexes in the reaction mixture.

Regrettably, the manuscript is inadequately prepared, featuring numerous factual errors in the crystallographic section. The manuscript, with editorial comments, has been submitted, yet recent changes are not apparent. There are concerns regarding the literature; while items have been altered in the text, it remains uncertain whether corresponding corrections have been made in the bibliography. The manuscript is challenging to comprehend. Furthermore, no literature references the CrysAlis Pro program, and the origin of the abbreviation LSM is unclear. 

Moreover, the illustrations are unclear and poorly presented, notably in Figure 1, Figure 3, and 4a. A significant issue is the CIF file in the supplement, which contains alerts A and B. This differs from the checkcif file in the supplement. Alerts A and B are disqualifying of crystallographic structures for publication. The values between atoms and bond lengths do not align with those presented in Tables S1 and S2. The authors lack knowledge of the notation for hydrogen bonding (D-H...A). Table 2 is poorly constructed and contains inaccuracies.

I firmly believe that the publication should not be accepted in its current state. It is poorly prepared and riddled with substantive errors that undermine its credibility.

Author Response

Regrettably, the manuscript is inadequately prepared, featuring numerous factual errors in the crystallographic section. – The manuscript has been heavily revised. We changed the title of the work, focusing on the study of the composition of Zr carboxylates formed in the ZrCl4 – 2-methylpropanoic acid system used for the preparation of efficient commercial catalysts for ethylene oligomerization.

There are concerns regarding the literature. - A number of references have been added to the revised manuscript. We would be grateful to the reviewer if he would draw our attention to possible errors in the list of references.

The manuscript is challenging to comprehend. – It is natural. Our work is not a crystallographic one. The aim of this work was to study the effect of the reaction conditions on the composition of Zr carboxylates formed in the ZrCl4 – isobutyric acid system used for the preparation of efficient commercial catalysts for ethylene oligomerization (alpha-Sablin).

No literature references the CrysAlis Pro program – CrysAlisPro data is in the cif file: CrysAlisPro, Agilent Technologies, Version 1.171.36.20 (release 27-06-2012 CrysAlis171 .NET) (compiled Jul 11 2012,15:38:31). We added the reference: Agilent Technologies (2012). CrysAlisPro, Version 1.171.36.20. AgilentTechnologies, Yarnton, Oxfordshire, England.

The origin of the abbreviation LSM is unclear. - The abbreviation LSM is a common abbreviation for the term “least squares method” and appears immediately after the term in the text. We have removed the abbreviation.

The illustrations are unclear and poorly presented, notably in Figure 1, Figure 3, and 4a. - Changes have been made to the Figures.

A significant issue is the CIF file in the supplement, which contains alerts A and B. This differs from the checkcif file in the supplement. Alerts A and B are disqualifying of crystallographic structures for publication. – We solved the structure in 2018, when CCDC requirements were not as strict as they are now. Checkcif file from that time was given in Supplementary. Now, the requirements have become more stringent. Note that we used X-ray diffraction only as an analytical tool. Taking into account the data of chemical analysis, IR spectroscopy and the close similarity of the known hexanuclear Zr 2-methylpropionate cluster and the considered Hf complex, we can say that we have determined the structure of the Hf cluster with an accuracy satisfactory for our purposes.

The values between atoms and bond lengths do not align with those presented in Tables S1 and S2. – We have made the appropriate corrections.

The authors lack knowledge of the notation for hydrogen bonding (D-H...A). - Table 2 is poorly constructed and contains inaccuracies. – Table 2 is taken from hydrogen bond data obtained using the SHELXL software package.

Round 2

Reviewer 1 Report

Comments and Suggestions for Authors

No further revision is needed.

Author Response

none

Reviewer 3 Report

Comments and Suggestions for Authors

Labels too big , in figures 1 and 3 and overlapping . choose sphere format for Hf atoms and pipe/wireframe for all other atoms.

In fig 2 only Hf need to be labelled.

re-refined CIF should be deposited with CCDC with structure factors , please confirm this has been done.

Author Response

Labels too big, in figures 1 and 3 and overlapping . choose sphere format for Hf atoms and pipe/wireframe for all other atoms – Figures are corrected.

In fig 2 only Hf need to be labelled. – It has been done.                                           

Re-refined CIF should be deposited with CCDC with structure factors , please confirm this has been done. – This has been done.

Reviewer 4 Report

Comments and Suggestions for Authors

The work has undoubtedly been improved and contains rational explanations of the research conducted.

Comments

line 220 Zr tetra carboxylate should be Zr(RCO2)4

line 230 should be crystals was determined.

line 261 Largest diff. peak and hole 3.317 and –3.214 e.Å –3 this should be comented in the text of MS.

line 336  "HfF3"?

line 404 "Table 2. Parameters of the hydrogen bonds in compound (IV)." should be Table 2. Parameters of the hydrogen bonds and weak contacts in compound (IV). HB with d(H..A) > 1.8 ?

Line 509 They did not succeed to reveal experimentally the difference
between μ3-O and μ3-OH groups. -

Localizing H atoms on oxygen atoms is a routine procedure and I doubt it will bring great scientific value to the discussion.

"Note that we used X-ray diffraction only as an analytical tool. Taking into account the data of chemical analysis, IR spectroscopy and the close similarity of the known hexanuclear Zr 2-methylpropionate cluster and the considered Hf complex, we can say that we have determined the structure of the Hf cluster with an accuracy satisfactory for our purposes. Difficulties in determining the structure are associated with the imperfection of the crystal and its tendency to crack. Our attempts to obtain a more perfect crystal have not yet led to success."

I also measure real crystals, not ideal ones, and the main alerts result from completely different things than the lack of a perfect crystal. Please refer to the literature and comment on the above state of affairs in at least one sentence. In other works, the authors had similar problems.

Author Response

Thank you for your thorough review. All comments have been taken into account in the revised text. The answers to the questions are given below.

Table 2. HB with d(H..A) > 1.8 – Table 2 presents results obtained when using the SHELXTL program package.

Localizing H atoms on oxygen atoms is a routine procedure and I doubt it will bring great scientific value to the discussion. – But not in the case of hexanuclear clusters. The authors of [14, 16, 17] did not reveal H atoms experimentally. That's why we highlight our results.

Largest diff. peak and hole 3.317 and –3.214 e.Å –3 this should be commented in the text. – This is commented.

Please refer to the literature and comment on the above state of affairs in at least one sentence. In other works, the authors had similar problems.  - There are immence reasons for larger than expected residual density maximum outside metal atom locations, such as errors on operating the diffractometer, the data processing software or the refinement software, temperature fluctuations at the detector, the crystal or at the X-ray source, cosmic rays, natural radioactivity, errors in absorption and extinction corrections, thermal diffuse scattering, radiation damage of the crystal, wrong scattering factors, e.g., nonrelativistic for heavy atoms , crystal chemical instability, twinning, wrongly assigned atom types, disorder, when the molecule under consideration is large and has bulky groups, etc.

Meindl, K.; Henn, J. Residual Density Analysis. Struct Bond (2012) 147: 143–192 Springer-Verlag Berlin Heidelberg 2010 Published online: 7 December 2010 DOI:10.1007/430_2010_26.

Comparing with literature data shows that our data on the largest peak-hole difference are consistent with published data for similar clusters (refs. 14, 16 in our manuscript).

  1. Table 2, entry 2.

Formula sum                                                              Zr12O70C144H264

Theta range for data collection                                 2.41–23.99

Largest difference in peak and hole, (e A_3)         3.922/-1.652

  1. Table 1, entry 3.

  Empirical formula                                            · C72H128O64Zr12· C18H36O12

  2θ range [°]                                                      1.46–23.26

  Largest diff. peak/hole [e·Å–3]                         3.211/–1.466

Table 1, entry 8.

Empirical formula                                           · C48H80Hf12O64·

2θ range [°]                                                     2.22–25.00

Largest diff. peak/hole [e·Å–3]                        4.319/–2.227

Our data

Empirical formula                                         C60H114Hf6O39

Theta range for data collection                    2.837 to 26.569 deg.

Reflections collected / unique                      38546 /17520

Largest diff. peak and hole                           3.317 and –3.214 e.Å–3

What concerns zirconium analog of our cluster, the authors [17] used the reflections data set obtained at noticeably smaller Θ angles when refining the structure:

Table 7

Empirical formula                                         C60H114O39Zr6

Theta range/deg                                          2.2–20.82

Reflections coll./unique                                16542/9412

Largest diff. peak and hole/e.A°                   0.823/ –0.783

The residual electron density is about 1. However, the data set used in the refinement was limited to the angle 2Θ=41 degrees. This limitation worsened the resolution, noticeably reduced the number of reflections while maintaining the number of refined parameters and made it possible to fit the model to the experiment. For verification, we also limited our experimental data set to the same angle 2Θ=41 degrees and obtained a similar result for the value of the residual electron density. However, we consider the use of such restrictions for our data set to be incorrect for the reasons stated above.

The problem of high residual density often arises when working with highly absorbing crystals. Sometimes the problem is solved if a small perfect crystal with the lowest possible linear absorption coefficient is selected for the experiment. This avoids the need to take absorption into account. But finding such a crystal is difficult and sometimes impractical if research plans do not include further study of such a substance.
